# Effect size, sample size and power of forced swim test assays in mice: Guidelines for investigators to optimize reproducibility

Neil R. Smalheiser[1]*, Elena E. Graetz[2], Zhou Yu[2], Jing Wang[2]

**1** Department of Psychiatry, University of Illinois School of Medicine, Chicago, Illinois, United States of America, **2** Department of Mathematics, Statistics, and Computer Science, University of Illinois at Chicago, Chicago, Illinois, United States of America

* neils@uic.edu

**Data Availability Statement:** All relevant data are within the manuscript and its Supporting Information files.

## Abstract

A recent flood of publications has documented serious problems in scientific reproducibility, power, and reporting of biomedical articles, yet scientists persist in their usual practices. Why? We examined a popular and important preclinical assay, the Forced Swim Test (FST) in mice used to test putative antidepressants. Whether the mice were assayed in a naïve state vs. in a model of depression or stress, and whether the mice were given test agents vs. known antidepressants regarded as positive controls, the mean effect sizes seen in the experiments were indeed extremely large (1.5–2.5 in Cohen's d units); most of the experiments utilized 7–10 animals per group which did have adequate power to reliably detect effects of this magnitude. We propose that this may at least partially explain why investigators using the FST do not perceive intuitively that their experimental designs fall short— even though proper prospective design would require ~21–26 animals per group to detect, at a minimum, large effects (0.8 in Cohen's d units) when the true effect of a test agent is unknown. Our data provide explicit parameters and guidance for investigators seeking to carry out prospective power estimation for the FST. More generally, altering the real-life behavior of scientists in planning their experiments may require developing educational tools that allow them to actively visualize the inter-relationships among effect size, sample size, statistical power, and replicability in a direct and intuitive manner.

## Introduction

A recent flood of publications has documented serious problems in scientific reproducibility, power, and reporting of biomedical articles, including psychology, neuroscience, and preclinical animal models of disease [1–16]. The power of published articles in many subfields of neuroscience and psychology hovers around 0.3–0.4, whereas the accepted standard is 0.8 [3, 4, 7, 9, 15]. Only a tiny percentage of biomedical articles specify prospective power estimations [e.g., 17]. This is important since under-powered studies not only produce excessive false-negative findings [18], but also have a tendency to over-estimate true effect sizes, and to show a

**Funding:** NS - This work was supported by National Institutes of Health Grants R01LM010817 https://www.nlm.nih.gov/ and P01AG039347 https://www.nia.nih.gov/. The funders had no role in study design, data collection and analysis, decision to publish, or preparation of the manuscript.

**Competing interests:** The authors have declared that no competing interests exist.

very high false-positive rate [1, 19]. Even when the nominal statistical significance of a finding achieves p = 0.05 or better, the possibility of reporting a false positive finding may approach 50% [1, 3, 20]. In several fields, when attempts have been made to repeat experiments as closely as possible, replication is only achieved about 50% of the time, suggesting that the theoretical critiques are actually not far from the real situation [6, 21].

Why might scientists persist in their usual practices, in the face of objective, clear evidence that their work collectively has limited reproducibility? Most critiques have focused on inadequate education or the incentives that scientists have to perpetuate the status quo. Simply put, scientists are instructed in "usual practice" and rewarded, directly and indirectly, for doing so [2, 3, 16]. There are more subtle reasons too; for example, PIs may worry that choosing an adequate number of animals per experimental group as specified by power estimation, if more than the 8–10 typically used in the field, will create problems in animal care committees who are concerned about reducing overall use of animals in research [22]. However, one of the major factors that causes resistance to change may be that investigators honestly do not have the perception that their own findings lack reproducibility [23].

In order to get a more detailed understanding of the current situation of biomedical experiments, particularly in behavioral neuroscience, we decided to focus on a single, popular and important preclinical assay, the Forced Swim Test (FST), which has been widely used to screen antidepressants developed as treatments in humans. Proper design of preclinical assays is important because they are used as the basis for translating new treatments to humans [eg., 22, 24]. Recently, Kara et al. presented a systematic review and meta-analysis of known antidepressants injected acutely in adult male mice, and reported extremely large mean effect sizes (Cohen's d ranging from 1.6 to 3.0 units) [25]. In this context, effect size refers to the difference in mean immobility time between treated and untreated groups in the FST assay, and conversion to Cohen's d units involves normalizing the effects relative to the standard deviation of responses across individuals of the same group. However, such antidepressants may have been originally chosen for clinical development (at least in part) because of their impressive results in the FST. Thus, in the present study, we have repeated and extended their analysis: making an unbiased random sampling of the FST literature, considering as separate cases whether the mice were assayed in a naïve state vs. in a model of depression or stress, and whether the mice were given test agents vs. known clinically prescribed antidepressants regarded as positive controls.

Our findings demonstrate that the mean effect sizes seen in the experiments were indeed extremely large; most of the experiments analyzed did have adequate sample sizes (defined as the number of animals in each group) and did have the power to detect effects of this magnitude. Our data go further to provide explicit guidelines for investigators planning new experiments using the Forced Swim Test, who wish to ensure that they will have adequate power and reproducibility when new, unknown agents are tested. We also suggest the need to develop tools that may help educate scientists to perceive more directly the relationships among effect size, sample size, statistical power (the probability that an effect of a given specified size will achieve statistical significance), and replicability (the probability that an experiment achieving statistical significance will, if repeated exactly, achieve statistical significance again).

## Materials and methods

In this study, searching PubMed using the query ["mice" AND "forced swim test" AND "2014/08/03"[PDat]: "2019/08/01"[PDat]] resulted in 737 articles, of which 40 articles were chosen at random using a random number generator. We only scored articles describing assays in which some test agent(s), e.g. drugs or natural products, postulated to have antidepressant properties, were given to mice relative to some control or baseline. Treatments might either be acute or

repeated, for up to 28 days prior to testing. Assays involving both male and female mice were included. Articles were excluded if they did not utilize the most common definition of forced swim test measures (i.e., the mice is in a tank for six minutes and during the last four minutes, the duration of immobility is recorded in seconds). We further excluded assays in rats or other species; assays that did not examine test agents (e.g. FST assays seeking to directly compare genetically modified vs. wild-type mice, or comparing males vs. females); interactional assays (i.e., assays to see if agent X blocks the effects of agent Y); and a few studies with extremely complex designs. When more than one FST assay satisfying the criteria was reported in a paper, all assays included were recorded and analyzed. We thus scored a total of 77 assays across 16 articles (S1 File).

Mean values and standard error were extracted from online versions of the articles by examining graphs, figures legends, and data in text if available. In addition, sample size, p-values and significance level were recorded. When sample size was not provided directly, it was inferred from t-test or ANOVA parameters and divided equally among treatment and groups, rounding up to the nearest whole number if necessary. If only a range for sample size was provided, the average of the range was assigned to all treatments, and rounded up if needed.

Control baseline immobility times were documented, indicating whether naïve mice were used or mice subjected to a model of depression or stress. To normalize effect size across experiments, Cohen's d was used since it is the most widely used measure [26, 27].

## Results

As shown in Table 1, across all assays, the FST effect sizes of both test agents and known clinically prescribed antidepressants regarded as positive controls had mean values in Cohen's d units of -1.67 (95% Confidence Interval: -2.12 to -1.23) and -2.45 (95% CI: -3.34 to -1.55), respectively. (Although Cohen's d units are defined as positive values, we add negative signs here to indicate that immobility times decreased relative to control values.) These are extremely large effects—twice as large as the standard definition of a "large" effect, i.e. a Cohen's d value of -0.8 [26, 27]!

The effect sizes of test agents vs. clinically prescribed antidepressants across all assays were not significantly different (two-tailed t-test for difference of means: t = 1.5859, p-value = 0.1202; Wilcoxon rank sum test for difference of medians: W = 839, p-value = 0.1347). We found no evidence for either ceiling or floor effects in these assays, that is, in no case did immobility times approach the theoretical minimum or maximum. The sample sizes (i.e., number of animals per treatment group) averaged 8–9 (Table 2).

### Assays in naïve mice vs. in models of depression or stress

Agents were tested for antidepressant effects in both naïve mice and mice subjected to various models of depression or stress. To our surprise, although one might expect longer baseline immobility times in "depressed" mice, our data indicate that the mean baseline immobility

**Table 1. Test agents vs. known antidepressants: Effect sizes.**

|  | MEAN | MEDIAN | SD | RANGE | CV |
|---|---|---|---|---|---|
| **TEST AGENTS N = 48** | -1.671 | -1.571 | 1.534 | -8.471, 0.759 | 0.918 |
| **ANTIDEPRESSANTS N = 29** | -2.448 | -2.144 | 2.354 | -9.428, 1.702 | 0.961 |

Shown are effect sizes (in Cohen's d units) for all FST assays that examined test agents and those that examined known clinically prescribed antidepressants regarded as positive controls (regardless of whether the effects achieved statistical significance). The mean effect size, median, range, and coefficient of variation (CV) are shown. The negative signs serve as a reminder that immobility times decreased relative to control values. N refers to the number of assays measured for each category.

**Table 2. Test agents vs. known antidepressants: Sample sizes.**

|  | MEAN | MEDIAN | SD | RANGE |
|---|---|---|---|---|
| **TEST AGENTS N = 48** | 8.31 | 8 | 2.183 | 6, 15 |
| **ANTIDEPRESSANTS N = 29** | 9.12 | 8 | 3.821 | 6, 24 |

Shown are sample sizes (number of animals per treatment group) for FST assays that examined test agents and those that examined known clinically prescribed antidepressants regarded as positive controls.

times of naïve and "depressed" mice (Fig 1, Table 3) did not differ significantly (one tailed t-test: p-value = 0.3375).

We then examined the effect sizes of test agents in naïve vs. depressive models (Table 4). There were no significant differences in mean effect size for test agents in naïve vs. depressed mice (two-tailed t-test t = -0.61513, p-value = 0.5423). Interestingly, the test agent assays in depressed models showed a smaller coefficient of variation (i.e., standard deviation divided by the mean) than in naïve mice. A smaller coefficient of variation in depressed models means that they show less intrinsic variability, which in turn means that it is easier for a given effect size to achieve statistical significance. (The number of assays of known antidepressants in depressed mice (N = 3) was too small in our sample to compare coefficients of variation vs. naïve mice.)

## Reporting parameters

None of the 16 randomly chosen articles in our dataset mentioned whether the FST assay was blinded to the group identity of the mouse being tested (although some did use automated

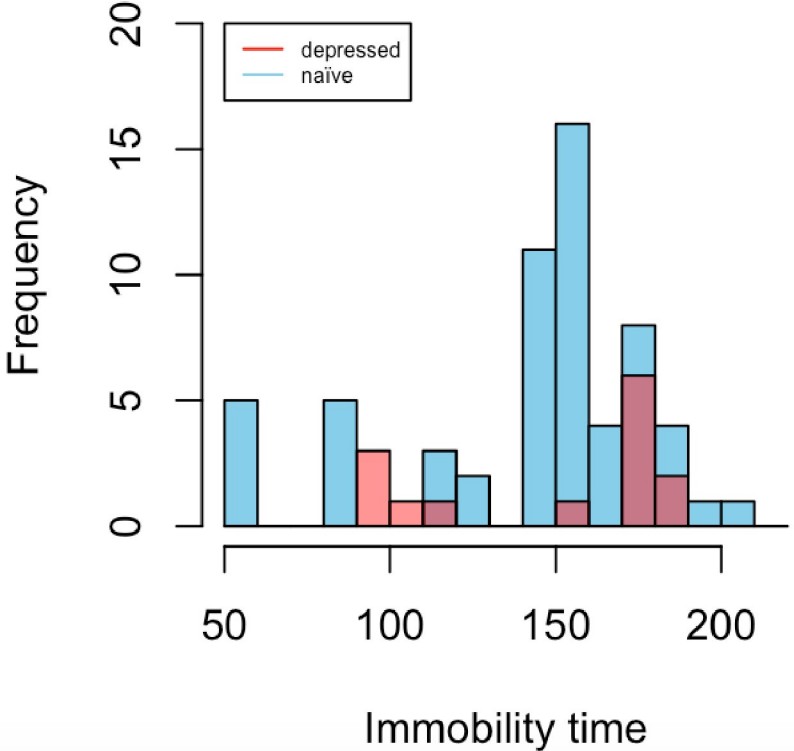

**Fig 1. Mean immobility times of control groups carried out on naïve mice vs. depressive models (same data as in Table 3).**

**Table 3. Control baseline immobility times in seconds.**

|  | MEAN | MEDIAN | SD | RANGE |
|---|---|---|---|---|
| **NAÏVE N = 63** | 143.817 | 159 | 38.985 | 56, 208 |
| **DEPRESSED N = 14** | 148.643 | 175 | 36.923 | 93, 184 |

systems to score the mice). None presented the raw data (immobility times) for individual mice. None discussed data issues such as removal of outliers, or whether the observed distribution of immobility times across animals in the same group was approximately normal or skewed. Only one mentioned power estimation at all (though no details or parameters were given). All studies utilized parametric statistical tests (t-test or ANOVA), which were either two-tailed or unspecified—none specified explicitly that they were using a one-tailed test.

## Discussion

Our literature analysis of the Forced Swim Test in mice agrees with, and extends, the previous meta-analysis of Kara et al [25], which found that known antidepressants exhibit extremely large effect sizes across a variety of individual drugs and mouse strains. We randomly sampled 40 recent articles of which 16 articles satisfied our basic criteria, comprising 77 antidepressant assays in which some test agent(s) were given to mice utilizing the most common definition of forced swim test measures). The mean FST effect sizes of both test agents and known clinically prescribed antidepressants regarded as positive controls had values in Cohen's d units of -1.67 (95% Confidence Interval: -2.12 to -1.23) and -2.45 (95% CI: -3.34 to -1.55), respectively. The 95% Confidence Intervals indicate that our sampling is adequate to support our conclusion, namely, that mean effect sizes are extremely large—and not anywhere near to the Cohen's d value of 0.8 which is generally thought of as a "large" mean effect size.

The first question that might be asked is whether the observed effects might be tainted by publication bias, i.e., if negative or unimpressive results were less likely to be published [10]. Ramos-Hryb et al. failed to find evidence for publication bias in FST studies of imipramine [28]. We cannot rule out bias against publishing negative results in the case of FST studies of test agents (i.e. agents not already clinically prescribed as antidepressants in humans), since nearly all articles concerning test agents reported positive statistically significant results (though not every assay in every article was significant). On the other hand, most if not all of the agents tested were not chosen at random, but had preliminary or indirect (e.g., receptor binding) findings in favor of their hypothesis.

The immobility time measured by the FST may reflect a discontinuous yes/no behavioral decision by mice, rather than a continuous variable like running speed or spontaneous activity. Kara et al [25] observed that the FST test does not exhibit clear dose-response curves in most of the published experiments that looked for them, which further suggests a switch-like rather

**Table 4. Test agents and known antidepressants in naïve vs. depressed models: Effect sizes.**

|  |  | MEAN | MEDIAN | SD | RANGE | CV |
|---|---|---|---|---|---|---|
| **TEST AGENTS** | Naïve N = 37 | -1.729 | -1.731 | 1.717 | -8.471, 0.759 | 0.993 |
|  | Depressed N = 11 | -1.496 | -1.231 | 0.826 | -3.406, -0.557 | 0.552 |
| **ANTIDEPRESSANTS** | Naïve N = 26 | -2.554 | -2.389 | 2.492 | -9.428, 1.702 | 0.975 |
|  | Depressed N = 3 | -2.115 | -0.856 | 2.255 | -4.718, -0.771 | 1.066 |

Shown are effect sizes (in Cohen's d units) for FST assays that examined test agents and those that examined known clinically prescribed antidepressants, in naïve or depressed models, respectively.

than graded response of the mice. This phenomenon may partially explain why effects in the FST appear to be very large and robust, and it complicates efforts to assess whether the effect sizes reported in the literature are inflated due to positive publication bias or low statistical power.

Surprisingly, we found that the baseline immobility time of naïve mice was not significantly different than the baseline immobility time of mice subjected to various models of depression or chronic stress (Table 2). This might potentially be explained by high variability of baseline values across heterogeneous laboratories and experimental variables such as strain, age, and gender. Alternatively, naïve mice housed and handled under routine conditions may be some-what "depressed" insofar as they have longer immobility times relative to those housed in more naturalistic environments [29].

## Guidelines for investigators using FST assays

One of the reasons that investigators rarely calculate prospective power estimations is the difficulty in ascertaining the necessary parameters accurately. Our results provide explicit values for these parameters for the FST, at least for the simple designs that are represented in our dataset. For example, for two independent groups of mice treated with an unknown test agent vs. control, one needs to enter a) the baseline immobility time expected in the control group (Table 3), b) the expected immobility time for the treated group (at the minimum biologically meaningful effect size that the investigator wishes to detect), c) the standard deviations of each group (Table 1), and d) the relative number of animals in each group (generally 1:1). Alternatively, one can enter the minimum biologically relevant effect size in Cohen's d units that the investigator wants to be able to detect (this encompasses both the difference in immobility times in the two groups as well as their standard deviations) (Table 5). This is sufficient to estimate the required number of animals per group (Table 5), assuming two groups (treated vs. control), standard criteria of power = 0.8, false-positive rate = 0.05, and a parametric statistical test (t-test or ANOVA). If the investigator has carried out preliminary (pilot) studies with a small number of experimental animals, that could represent an useful alternative in order to calculate the experimental number needed to achieve the desired power.

## But the power of current FST assays is adequate, isn't it?

From Tables 1 and 4, one can see that the observed mean effect sizes across the literature fall into the range of 1.5 to 2.5 Cohen's d units and for the sake of this discussion, we will assume that these values are not inflated. Indeed, if an investigator merely wants to be able to detect effects of this size, only 7–8 animals per group are required, which is in line with the number actually used in these experiments (Table 5). This is likely to explain why scientists in this field

**Table 5. Prospective power estimation for test agents in the FST assay.**

|  | EFFECT SIZE | #ANIMALS REQUIRED PER GROUP |
|---|---|---|
| MODERATE ES | -0.5 | 64 |
| LARGE ES | -0.8 | 26 |
| MEAN ES (THIS STUDY) | -1.671 | 7 |
| MEDIAN ES (THIS STUDY) | -1.572 | 7 |

These sample size calculations are based on the observed mean and median effect sizes (ES) in Cohen's d units for novel test agents (Table 1), two groups (treated vs. controls), for desired power = 0.8, alpha = 0.05, and two-sided t-test or ANOVA [25].

have the intuition that the empirical standard sample size of 8–9 (Table 2) is enough to ensure adequate power.

However, setting the **minimum** effect size at the observed **mean (or median)** value is clearly not satisfactory since half of the assays fall below that value. When an investigator is examining an unknown test agent, the general guidance is to set the minimum effect size at "moderate" (0.5) if not "large" (0.8) [30], which would require 64 or 26 animals per group, respectively, in order to ensure adequate power (Table 5). Setting the minimum effect size is not something to be fixed, and depends not only on the assay but also on the investigator's hypothesis to be tested [31]. Nevertheless, the appropriate minimum should always be set smaller than the mean observed effect size of the assay as a whole, especially when the agent to be tested lacks preliminary evidence showing efficacy. From this perspective, a new FST experiment planned using 7–10 animals will be greatly under-powered. Nevertheless, this does shed light on why scientists performing the FST assay may not intuitively perceive that their experiments are under-powered.

## Possible experimental design strategies for improved power

**One tail or two?.**   Investigators in our dataset never stated that they used one-tailed statistical tests, even though they generally had preliminary or suggestive prior evidence suggesting that the agent being tested may have antidepressant effects in the FST. Using a one-tailed hypothesis in prospective power estimation reduces the number of animals needed per group, for the same power and false-positive rate. For a minimum effect size of 0.8, a two-tailed hypothesis that requires 26 animals per group reduces to 21 animals per group for a one-tailed hypothesis [32].

In summary, for testing an unknown agent (e.g., chosen without prior experimental evidence or as part of a high-throughput screen), with minimum effect size = 0.8, power = 0.8 and false-positive rate = 0.05, the results suggest that an investigator should use a two-tailed hypothesis and will need ~26 animals per group. (High throughput assays will need additional post hoc corrections for multiple testing.) For a test agent which has preliminary or prior evidence in favor of being an antidepressant, a one-tailed hypothesis is appropriate and ~21 animals per group can be used. Note that this discussion applies to simple experimental designs only. Interactional assays (e.g., does agent X block the effects of agent Y?) are expected to have larger standard deviations than direct assays and would require somewhat larger sample sizes, as would complex experimental designs of any type.

**Parametric or nonparametric testing?.**   All experiments in our dataset employed parametric statistical tests, either ANOVA or t-test. This is probably acceptable when sample sizes of 20 or more are employed, as recommended in the present paper, but not for the usual 7–10 animals per group, as performed by most of the investigators in our dataset. This is for two reasons:

First, investigators in our dataset have not presented the raw data for individual animals in each group to verify that the underlying data distribution across individuals resembles a normal distribution. If indeed immobility responses have a switch-like aspect (see above), one might expect that responses across individuals might tend to be bimodal or skewed. In the absence of individual-level data, we plotted the mean effect sizes for known antidepressants plotted across the different assays (Fig 2) and this distribution passes the Shapiro Wilk test for normality (p-value = 0.231).

Second, when sample sizes are so small, parametric tests have a tendency to ascribe too much significance to a finding [14], and together with the issue of inflated effect sizes, this results in over-optimistic prospective power estimation. Nonparametric tests such as the Wilcoxon signed rank test (with either one-tailed or two-tailed hypothesis) are appropriate

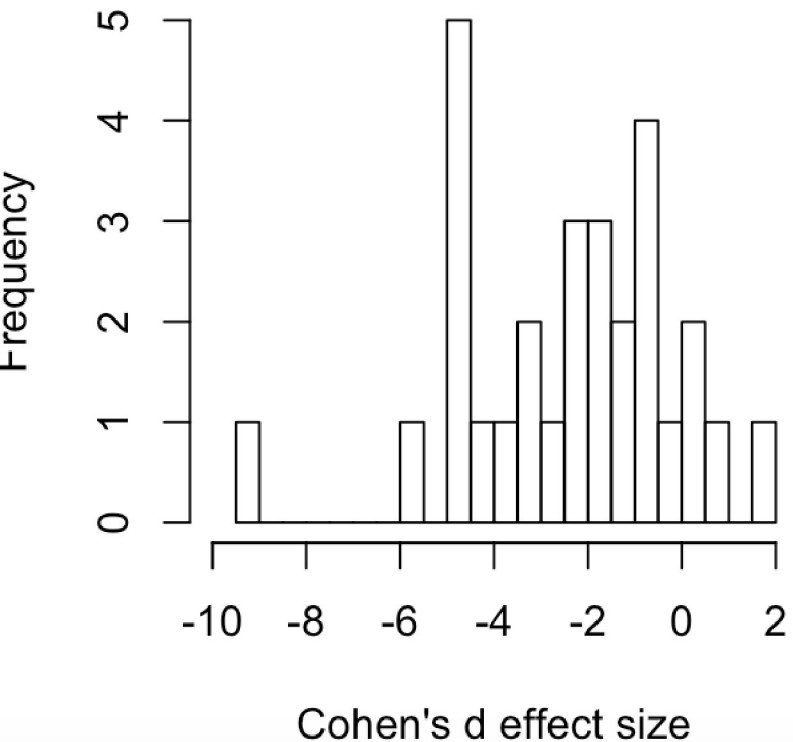

**Fig 2. Effect sizes of known antidepressants across all assays (N = 29; see Table 1).**

regardless of normality, and will be more conservative than parametric tests, i.e. will have less tendency to ascribe too much significance to a finding [14]. Popular software including G*Power are able to handle nonparametric testing [32]. A warning though: Using a nonparametric test will result in estimates of required sample sizes larger than those obtained using parametric tests.

**Within-animal design?.**   None of the assays in our dataset involved a before/after design in the same animals. This means giving a control vs. an agent to a mouse, observing the immobility time in the FST assay, then repeating the assay in the same mouse with the other treatment. Using an individual mouse as its own control has the advantage of less variability (i.e. no inter-animal variability needs to be considered) and allows the investigator to use paired statistics instead of unpaired tests. Both of these advantages should tend to increase power for the same number of animals, plus, one can divide the number of total animals needed in half since each one is its own control. Unfortunately, control baseline immobility times are not stable on retesting, and investigators have found that the test-retest scheme results in similar effect sizes as the standard assay in some but not all cases [26, 33–35]. Thus, one would need to employ test-retest FST paradigms with some caution and with extra controls.

## Limitations of our study

Our literature analysis did not examine how effect sizes may vary across mouse strain, age, gender, or across individual drugs [25]. Because the number of animals used in each experimental group was often not explicitly given, we imputed sample sizes from tabular legends or t-test or ANOVA parameters in some cases. We also did not undertake a Bayesian analysis to estimate the prior probability that any given test agent chosen at random will have

antidepressant effects in the FST assay. We did not consider how power might be affected if animals are not truly independent (e.g. they may be littermates) and if they are not randomly allocated to groups [36]. Our guidelines do not encompass designs in which the sample size is not pre-set at the outset [37]. As well, we did not directly assess the replicability of published FST experiments, i.e., if one publication reports a statistically significant finding, what is the probability that another group examining the same question will also report that the finding is statistically significant? Replicability is related to adequate statistical power but also involves multiple aspects of experimental design not considered here [2, 5, 8, 11, 13, 20, 38]. Nevertheless, adequate power is essential for experiments to be replicable, because under-powered studies tend to produce high false-negative findings [18], to over-estimate effect sizes, and to have inflated false-positive rates [4, 39].

Finally, it must be acknowledged that none of the preclinical antidepressant assays carried out in animals fully reproduce all aspects of depression pathophysiology or treatment response in humans [40, 41]. Therefore, regardless of effect sizes or reproducibility of animal findings, one must make a conceptual leap when considering the clinical promise of any given antidepressant drug.

## Conclusions

In the case of the Forced Swim Test used to assess antidepressant actions of test agents in mice, we found that the mean effect size is extremely large (i.e., 1.5–2.5 in Cohen's d units), so large that only 7–10 animals per group are needed to reliably detect a difference from controls. This may shed light on why scientists in neuroscience, and preclinical biomedical research in general, have the intuition that their usual practice (7–10 animals per group) provides adequate statistical power, when many meta-science studies have shown that the overall field is greatly under-powered. The large mean effect size may at least partially explain why investigators using the FST do not perceive intuitively that their experimental designs fall short. It can be argued that when effects are so large, relatively small sample sizes may be acceptable [42]. The Forced Swim Test is not unique–to name one example, rodent fear conditioning is another popular preclinical assay that exhibits extremely large effect sizes [43]. Nevertheless, we showed that adequate power to detect minimum biologically relevant large effects in this assay actually requires at least ~21–26 animals per group when the true effect of a test agent is unknown.

We suggest that investigators are not able to perceive intuitively whether or not a given sample size is adequate for a given experiment, and this contributes to a mindset that is skeptical of theoretical or statistical arguments. Apart from other educational and institutional reforms [2, 3, 10, 11, 13, 20, 22, 38, 44], altering the real-life behavior of scientists in planning their experiments may require developing tools that allow them to actively visualize the interrelationships among effect size, sample size, statistical power, and replicability in a direct and intuitive manner.

## Supporting information

**S1 File. Excel spreadsheet of the FST assays scored in this paper.** A total of 77 FST assays across 16 randomly chosen articles were scored. See text for details.
(XLSX)

## Author Contributions

**Conceptualization:** Neil R. Smalheiser.

**Data curation:** Neil R. Smalheiser, Elena E. Graetz, Zhou Yu.

**Formal analysis:** Elena E. Graetz.

**Funding acquisition:** Neil R. Smalheiser.

**Investigation:** Neil R. Smalheiser, Elena E. Graetz, Zhou Yu.

**Methodology:** Neil R. Smalheiser.

**Project administration:** Neil R. Smalheiser.

**Supervision:** Neil R. Smalheiser, Jing Wang.

**Writing – original draft:** Neil R. Smalheiser.

**Writing – review & editing:** Neil R. Smalheiser, Elena E. Graetz, Zhou Yu, Jing Wang.

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
