## [Decision Letter · Decision Letter 0]

20 Jan 2021

PONE-D-20-36904

Effect size, sample size and power of forced swim test assays in mice: Guidelines for investigators to optimize reproducibility

PLOS ONE

Dear Dr. Smalheiser,

Thank you for submitting your manuscript to PLOS ONE. After careful consideration, we feel that it has merit but does not fully meet PLOS ONE’s publication criteria as it currently stands. Therefore, we invite you to submit a revised version of the manuscript that addresses the points raised during the review process.

We look forward to receiving your revised manuscript.

Kind regards,

Patricia Souza Brocardo, Ph.D.

Academic Editor

PLOS ONE

Reviewers' comments:

Reviewer's Responses to Questions

**Comments to the Author**

1. Is the manuscript technically sound, and do the data support the conclusions?

Reviewer #1: Yes

Reviewer #2: Yes

2. Has the statistical analysis been performed appropriately and rigorously? 

Reviewer #1: Yes

Reviewer #2: Yes

3. Have the authors made all data underlying the findings in their manuscript fully available?

Reviewer #1: Yes

Reviewer #2: Yes

4. Is the manuscript presented in an intelligible fashion and written in standard English?

Reviewer #1: No

Reviewer #2: Yes

5. Review Comments to the Author

Reviewer #1: With a particular emphasis on the relationship between sample size, statistical power and effect size, the authors explored a sample of published research articles in order to shed light into these topics, as well as to discuss reproducibility and the behavior of scientists in planning their experiments. The sample (16 published articles) was selected from a population of mouse-based experimental studies aimed to investigate potential antidepressant molecules in the Forced Swim Test (FST), which has been used to test putative antidepressants. In brief, the authors observed extremely large mean effect sizes in the analyzed sample (1.5 – 2.5 in Cohen’s d units). The average experimental number per group (7-10 animals), although relatively small, was considered enough to achieve adequate power to consistently detect effects of this magnitude (1.5 – 2.5 in Cohen’s d units). In order to detect large effects (0.8 in Cohen’s d units) when the true effect of a test agent is unknown, the authors propose that proper prospective design would require approximately 21-26 animals per group, which is a number significantly higher than the current average (7-10 animals). The authors discuss their data emphasizing that they provide parameters and guidance for investigators seeking to carry out prospective power estimation for the FST. They also state that the investigators are not able to perceive intuitively whether or not a given sample size is adequate for a given experiment, which seems to be related, at least partially, with the extremely large mean effect sizes detected (1.5 – 2.5 in Cohen’s d units).

The manuscript, which is properly written and presented, brings out an interesting view on the current scenario of FST, search for antidepressants, sample size and statistical power. However, there are points that deserve major consideration in order to improve the manuscript:

1 - In some parts of the manuscript (lines 46 and 327), the authors argue that underpowered studies have a tendency to show a very high false-positive rate. It is important to present information on the (also high) false-negative rate in underpowered studies, particularly in studies with small (but significant) effects.

2 - Does a 16-articles sample have enough power to properly represent the population of 737 articles? Based on statistical foundation, the authors need to discuss this topic in order to inform the reader about the reliability of their data, which derived from a relatively small sample (n = 16).

3 - In the Materials and Methods Section, line 107, it is informed “When sample size was not provided directly, it was inferred from t-test or ANOVA parameters and divided equally among treatment and groups, rounding up to the nearest whole number if necessary. If only a range for sample size was provided, the average of the range was assigned to all treatments, and rounded up if needed”. This also represents a limitation of the study because there are a huge number of articles with different experimental number in their different experimental groups. This should, at least, be informed in the limitations of the study (line 314).

4 - Line 128: Was the effect size of known antidepressants normally distributed in the sample (16 articles)? Please, add more on that.

5 - It would be informative to show data concerning the dispersion of immobility-time within and between studies, at least for controls.

6 - Line 169: “Interestingly, the assays in depressed models showed a smaller coefficient of variation (i.e., standard deviation divided by the mean) than in naïve mice”. This seems to be true only for "test agents", but not for "antidepressants". Please, revise the sentence. In addition, discuss it.

7 - Table 4: With respect to “antidepressants/naïve”, the mean effect size of 2.554 is considered high. However, several “known antidepressants" have no clinical effects in a significant number of patients (refractory). Even though 1.729 (“test agents/naïve”) is also considered a high mean effect size, is it possible to predict a worse clinic effect (in patients) compared to “antidepressants/naïve” (effect size 2.554)? In my opinion, discussions concerning the biological relevance of the data found in the evaluated articles would improve the present manuscript relevance.

8 - Line 220: “This might potentially be explained by high variability of baseline values across heterogeneous experiments and laboratories”. This is not shown. Please, see comment 5.

9 - Line 276: “In summary, for testing an unknown agent (e.g., chosen without prior experimental evidence or as part of a high-throughput screen), with minimum effect size = 0.8, power = 0.8 and false positive rate = 0.05, the results suggest that an investigator should use a two-tailed hypothesis and will need ~26 animals per group”. Taking into account that the present manuscript intents to provide guidance for investigators, it would be important to inform the readers that the development of preliminary (pilot) studies, with small experimental number, could represent an useful alternative in order to calculate the experimental number needed to achieve the desired power.

Reviewer #2: General Comments to the Authors:

This is a well thought-out and designed study that provides some clear guidelines for researchers and scientists on how to optimize behavioural studies (using the Forced Swim Test as an example). It highlights the importance of statistical considerations when designing a study and particularly deciding on the number of animals that should be included in the analysis to ensure reproducibility. I believe this to be a very important study in shifting some of the engrained beliefs around the adequate power to detect reliable effects. At this moment, I only have a few specific comments/suggestions for the authors to consider before the manuscript can be accepted for publication.

Specific Comments to the Authors:

1. Introduction: It would be great if the Authors could briefly define these Terms the first time they are referred to in the manuscript: effect size, sample size, statistical power, and replicability. This will improve the paper's readability and will ensure that the reader understands the differences between effect size and sample size, for example.

2. Discussion, page 9, lines 210-216: According with this hypothesis that the FST may reflect a discontinuous YES/NO behavioural decision by mice, would it then be more appropriate to use non-parametric statistical tests when analyzing FST data? Can the Authors comment/discuss this in the Discussion of the Manuscript (by perhaps tying this in with the Discussion on “Parametric or Nonparametric Testing” on page 12)?

3. Discussion, page 9, lines 218-223: Other potential variables that may account for the variability in baseline include: strain of mice used and age of the animals at the time of the experiment (i.e., comparing baseline of young animals vs. old animals, for example). These additional variables should also be listed and discussed here.

4. Discussion: Can the Authors comment/provide practical suggestions on how the current body of literature (i.e., published studies using the FST to test the putative antidepressant effects of drugs in mice) should be interpreted in light of the findings reported in this study?

6. PLOS authors have the option to publish the peer review history of their article (what does this mean?). If published, this will include your full peer review and any attached files.

Reviewer #1: **Yes: **Marcelo Farina

Reviewer #2: No

---

## [Author Response · Author response to Decision Letter 0]

27 Jan 2021

Reviewer #1: With a particular emphasis on the relationship between sample size, statistical power and effect size, the authors explored a sample of published research articles in order to shed light into these topics, as well as to discuss reproducibility and the behavior of scientists in planning their experiments. The sample (16 published articles) was selected from a population of mouse-based experimental studies aimed to investigate potential antidepressant molecules in the Forced Swim Test (FST), which has been used to test putative antidepressants. In brief, the authors observed extremely large mean effect sizes in the analyzed sample (1.5 – 2.5 in Cohen’s d units). The average experimental number per group (7-10 animals), although relatively small, was considered enough to achieve adequate power to consistently detect effects of this magnitude (1.5 – 2.5 in Cohen’s d units). In order to detect large effects (0.8 in Cohen’s d units) when the true effect of a test agent is unknown, the authors propose that proper prospective design would require approximately 21-26 animals per group, which is a number significantly higher than the current average (7-10 animals). The authors discuss their data emphasizing that they provide parameters and guidance for investigators seeking to carry out prospective power estimation for the FST. They also state that the investigators are not able to perceive intuitively whether or not a given sample size is adequate for a given experiment, which seems to be related, at least partially, with the extremely large mean effect sizes detected (1.5 – 2.5 in Cohen’s d units).

The manuscript, which is properly written and presented, brings out an interesting view on the current scenario of FST, search for antidepressants, sample size and statistical power. However, there are points that deserve major consideration in order to improve the manuscript:

1 - In some parts of the manuscript (lines 46 and 327), the authors argue that underpowered studies have a tendency to show a very high false-positive rate. It is important to present information on the (also high) false-negative rate in underpowered studies, particularly in studies with small (but significant) effects.

 Thank you for this comment. We have now mentioned the false-negative issue and added a reference, Fiedler K, Kutzner F, Krueger JI. The Long Way From α-Error Control to Validity Proper: Problems With a Short-Sighted False-Positive Debate. Perspect Psychol Sci. 2012 Nov;7(6):661-9. doi: 10.1177/1745691612462587. 

2 - Does a 16-articles sample have enough power to properly represent the population of 737 articles? Based on statistical foundation, the authors need to discuss this topic in order to inform the reader about the reliability of their data, which derived from a relatively small sample (n = 16).

 Thank you for this comment. We did address this issue implicitly by presenting the 95% confidence interval for effect sizes – even extracting “only” 77 assays in 16 articles, the effect sizes were extremely large: “the mean FST effect sizes of both test agents and known clinically prescribed antidepressants regarded as positive controls had values in Cohen’s d units of -1.67 (95% Confidence Interval: -2.12 to -1.23) and -2.45 (95% CI: -3.34 to -1.55), respectively.” In the revised version, we add some language in the Discussion: “95% Confidence Intervals indicate that our sampling is adequate to support our conclusion, namely, that mean effect sizes are extremely large -- and not anywhere near to the Cohen’s d value of 0.8 which is generally thought of as a “large” effect size.”

3 - In the Materials and Methods Section, line 107, it is informed “When sample size was not provided directly, it was inferred from t-test or ANOVA parameters and divided equally among treatment and groups, rounding up to the nearest whole number if necessary. If only a range for sample size was provided, the average of the range was assigned to all treatments, and rounded up if needed”. This also represents a limitation of the study because there are a huge number of articles with different experimental number in their different experimental groups. This should, at least, be informed in the limitations of the study (line 314).

 Thanks, we have now added this as a limitation. 

4 - Line 128: Was the effect size of known antidepressants normally distributed in the sample (16 articles)? Please, add more on that.

 In the revised version, we have now added a histogram and a test of normality for this distribution. 

5 - It would be informative to show data concerning the dispersion of immobility-time within and between studies, at least for controls.

 The reports did not provide any data on the dispersion within studies, but in the revised version, we have now added a histogram on the immobility times across studies. 

6 - Line 169: “Interestingly, the assays in depressed models showed a smaller coefficient of variation (i.e., standard deviation divided by the mean) than in naïve mice”. This seems to be true only for "test agents", but not for "antidepressants". Please, revise the sentence. In addition, discuss it.

 We have now revised the sentence and we now explain that: “The number of assays of known antidepressants in depressed mice (N = 3) was too small in our sample to compare coefficients of variation vs. naïve mice.”

7 - Table 4: With respect to “antidepressants/naïve”, the mean effect size of 2.554 is considered high. However, several “known antidepressants" have no clinical effects in a significant number of patients (refractory). Even though 1.729 (“test agents/naïve”) is also considered a high mean effect size, is it possible to predict a worse clinic effect (in patients) compared to “antidepressants/naïve” (effect size 2.554)? In my opinion, discussions concerning the biological relevance of the data found in the evaluated articles would improve the present manuscript relevance.

 The reviewer raises a set of profound questions about the relationship of animal assay responses to human depressive treatment response. Our data do not directly address these issues. However, in the revised ms., in the Limitations section, we have now added several references that point to this issue: “Finally, it must be acknowledged that none of the preclinical antidepressant assays carried out in animals fully reproduce all aspects of depression pathophysiology or treatment response in humans. Therefore, regardless of effect sizes or reproducibility of animal findings, one must make a conceptual leap when considering the clinical promise of any given antidepressant drug.”

8 - Line 220: “This might potentially be explained by high variability of baseline values across heterogeneous experiments and laboratories”. This is not shown. Please, see comment 5.

 Added a histogram as mentioned above. 

9 - Line 276: “In summary, for testing an unknown agent (e.g., chosen without prior experimental evidence or as part of a high-throughput screen), with minimum effect size = 0.8, power = 0.8 and false positive rate = 0.05, the results suggest that an investigator should use a two-tailed hypothesis and will need ~26 animals per group”. Taking into account that the present manuscript intents to provide guidance for investigators, it would be important to inform the readers that the development of preliminary (pilot) studies, with small experimental number, could represent an useful alternative in order to calculate the experimental number needed to achieve the desired power.

 Done. 

Reviewer #2: General Comments to the Authors:

This is a well thought-out and designed study that provides some clear guidelines for researchers and scientists on how to optimize behavioural studies (using the Forced Swim Test as an example). It highlights the importance of statistical considerations when designing a study and particularly deciding on the number of animals that should be included in the analysis to ensure reproducibility. I believe this to be a very important study in shifting some of the engrained beliefs around the adequate power to detect reliable effects. At this moment, I only have a few specific comments/suggestions for the authors to consider before the manuscript can be accepted for publication.

Specific Comments to the Authors:

1. Introduction: It would be great if the Authors could briefly define these Terms the first time they are referred to in the manuscript: effect size, sample size, statistical power, and replicability. This will improve the paper's readability and will ensure that the reader understands the differences between effect size and sample size, for example.

 Thank you for your comment. We have now defined these terms in the Introduction.

2. Discussion, page 9, lines 210-216: According with this hypothesis that the FST may reflect a discontinuous YES/NO behavioural decision by mice, would it then be more appropriate to use non-parametric statistical tests when analyzing FST data? Can the Authors comment/discuss this in the Discussion of the Manuscript (by perhaps tying this in with the Discussion on “Parametric or Nonparametric Testing” on page 12)?

 We agree entirely! We now added the following sentence to the Discussion: “If indeed immobility responses have a switch-like aspect (see above), one might expect that responses might be bimodal or skewed.”

3. Discussion, page 9, lines 218-223: Other potential variables that may account for the variability in baseline include: strain of mice used and age of the animals at the time of the experiment (i.e., comparing baseline of young animals vs. old animals, for example). These additional variables should also be listed and discussed here.

 In the revised ms., we have now added age, strain and gender to the list here as well as in the Limitations to our analysis. 

4. Discussion: Can the Authors comment/provide practical suggestions on how the current body of literature (i.e., published studies using the FST to test the putative antidepressant effects of drugs in mice) should be interpreted in light of the findings reported in this study?

 The reviewer astutely notices that we have refrained from critiquing/discussing the way that the FST is used as a behavioral endpoint for animal neuroscience studies, nor the relevance of the FST as a predictor of clinical promise in humans. We also do not walk readers through the entire process of how to design FST experiments (which would involve how to choose informative agents to test in the first place, the best strains of mice, possible use of flexible stopping points, etc.). These subjects are important but are much too large for the present paper to consider. Rather, we deliberately restricted practical suggestions to a narrow scope: how experimenters could/should improve their existing designs in terms of sample size, statistics, etc. These considerations should be of somewhat general interest insofar as they generalize to other preclinical assays. In the revised ms., we have now added discussion and references (as mentioned above) that acknowledge the leap between preclinical antidepressant assays in animals and clinical antidepressant treatment in humans.

---

## [Decision Letter · Decision Letter 1]

10 Feb 2021

Effect size, sample size and power of forced swim test assays in mice:

Guidelines for investigators to optimize reproducibility

PONE-D-20-36904R1

Dear Dr. Smalheiser,

We’re pleased to inform you that your manuscript has been judged scientifically suitable for publication and will be formally accepted for publication once it meets all outstanding technical requirements.

Kind regards,

Patricia Souza Brocardo, Ph.D.

Academic Editor

PLOS ONE

Reviewers' comments:

Reviewer's Responses to Questions

**Comments to the Author**

1. If the authors have adequately addressed your comments raised in a previous round of review and you feel that this manuscript is now acceptable for publication, you may indicate that here to bypass the “Comments to the Author” section, enter your conflict of interest statement in the “Confidential to Editor” section, and submit your "Accept" recommendation.

Reviewer #1: All comments have been addressed

Reviewer #2: All comments have been addressed

2. Is the manuscript technically sound, and do the data support the conclusions?

Reviewer #1: Yes

Reviewer #2: Yes

3. Has the statistical analysis been performed appropriately and rigorously? 

Reviewer #1: Yes

Reviewer #2: Yes

4. Have the authors made all data underlying the findings in their manuscript fully available?

Reviewer #1: Yes

Reviewer #2: Yes

5. Is the manuscript presented in an intelligible fashion and written in standard English?

Reviewer #1: Yes

Reviewer #2: Yes

6. Review Comments to the Author

Reviewer #1: Based on reading of the revised version and authors' response, it is evident that the manuscript was properly revised and significantly improved.

Reviewer #2: All my previous comments have now been addressed and I do not have any further suggestions at this point. I am happy to now endorse this version of the manuscript for publication.

7. PLOS authors have the option to publish the peer review history of their article (what does this mean?). If published, this will include your full peer review and any attached files.

Reviewer #1: No

Reviewer #2: No

---

## [Editor Report · Acceptance letter]

15 Feb 2021

PONE-D-20-36904R1 

Effect size, sample size and power of forced swim test assays in mice: Guidelines for investigators to optimize reproducibility 

Dear Dr. Smalheiser:

I'm pleased to inform you that your manuscript has been deemed suitable for publication in PLOS ONE. Congratulations! Your manuscript is now with our production department. 

Kind regards, 

on behalf of

Dr. Patricia Souza Brocardo 

Academic Editor

PLOS ONE